# Phenolic Composition of *Crataegus monogyna* Jacq. Extract and Its Anti-Inflammatory, Hepatoprotective, and Antileukemia Effects

**DOI:** 10.3390/ph17060786

**Published:** 2024-06-15

**Authors:** Fatima Ez-Zahra Amrati, Ibrahim Mssillou, Smahane Boukhira, Mehdi Djiddi Bichara, Youness El Abdali, Renata Galvão de Azevedo, Chebaibi Mohamed, Meryem Slighoua, Raffaele Conte, Sotirios Kiokias, Gemilson Soares Pontes, Dalila Bousta

**Affiliations:** 1Laboratory of Cell Biology and Molecular Genetics (LBCGM), Department of Biology, Faculty of Sciences, Ibn Zohr University, Agadir 8106, Morocco; 2Laboratory of Natural Substances, Pharmacology, Environment, Modeling, Health & Quality of Life (SNAMOPEQ), Faculty of Sciences Dhar El Mahraz, Sidi Mohamed Ben Abdellah University, Fez 30000, Morocco; ibrahim.mssillou@usmba.ac.ma; 3Higher Institute of Nursing Professions and Health Techniques, Ministry of Health and Social Protection, Guelmim 81000, Morocco; smahane.boukhira@usmba.ac.ma; 4Laboratory of Biotechnology, Health, Agrofood and Environment (LBEAS), Faculty of Sciences Dhar El Mahraz, Sidi Mohamed Ben Abdellah University, Fez 30000, Morocco; djidibichara.mehdi@usmba.ac.ma (M.D.B.); youness.elabdali@usmba.ac.ma (Y.E.A.);; 5Laboratory of Virology, National Institute of Amazonian Research (INPA), Manaus 69067-375, Brazil; renata.azevedo@ufam.edu.br (R.G.d.A.);; 6Post-Graduate Program in Basic and Applied Immunology, Institute of Biological Science, Federal University of Amazonas, Manaus 69077-000, Brazil; 7Higher Institute of Nursing Professions and Health Techniques, Ministry of Health and Social Protection, Fez 30050, Morocco; 8Biomedical and Translational Research Laboratory, Faculty of Medicine and Pharmacy of the Fez, Sidi Mohamed Ben Abdellah University, Fez 30000, Morocco; 9Research Institute on Terrestrial Ecosystems (IRET), National Research Council, 05010 Naples, Italy; raffaele-conte@cnr.it; 10European Research Executive Agency (REA), 1210 Brussels, Belgium; sotirios.kiokias@ec.europa.eu; 11National Agency of Medicinal and Aromatic Plants, Taounate 34000, Morocco

**Keywords:** *Crataegus monogyna*, inflammation, hepatotoxicity, human hepatocellular carcinoma, leukemia, in silico studies

## Abstract

*Crataegus monogyna* (*C. monogyna*) is a prominent plant used in Moroccan traditional medicine. This study investigated the phenolic composition and the anti-inflammatory, the hepatoprotective, and the anticancer activities of a hydroethanolic extract of *C. monogyna* leaves and stems. Ultra-high-performance liquid chromatography identified the phenolic profile. The in vitro anticancer activity was evaluated using the MTT assay on HL-60 and K-562 myeloleukemia cells and liver (Huh-7) cell lines. The anti-inflammatory effect was assessed in vivo using carrageenan-induced paw edema in rats. The hepatoprotective effect at 300 and 1000 mg/kg doses against the acetaminophen-induced hepatotoxicity on rats was studied for seven days. Additionally, molecular docking simulations were performed to evaluate the extract’s inhibitory potential against key targets: lipoxygenase, cytochrome P450, tyrosine kinase, and TRADD. The extract exhibited significant cytotoxic activity against K-562 and HL-60 cells, but not against lung cancer cells (Huh-7 line). The 1000 mg/kg dose demonstrated the most potent anti-inflammatory effect, inhibiting edema by 99.10% after 6 h. *C. monogyna* extract displayed promising hepatoprotective properties. Procyanidin (−7.27 kcal/mol), quercetin (−8.102 kcal/mol), and catechin (−9.037 kcal/mol) were identified as the most active molecules against lipoxygenase, cytochrome P450, and tyrosine kinase, respectively. These findings highlight the untapped potential of *C. monogyna* for further exploration in treating liver damage, inflammation, and leukemia.

## 1. Introduction

Herbal treatments have been shown to serve as promising novel therapeutic agents. This is due to the vast array of medicinal plants with potentially bioactive compounds that can be explored for their effectiveness in treating various ailments [1,2]. *C. monogyna* (common hawthorn), a member of the Rosaceae family, is part of the flora in Morocco and is found in Europe, Asia, and North America [3]. Various species of Crataegus predominate in these regions, and different parts of the plant are utilized. *Crataegus monogyna* Jacq. is one of the most used species of this genus in the Moroccan traditional medicine. It has been used for several illnesses, especially to cure digestive system problems, microbial diseases, and diabetes [4].

Hawthorn is an excellent source of bioactive compounds, including various polyphenols like chlorogenic acid, procyanidins, isoquercitrin, and epicatechin. It also contains several triterpenic acids, such as oleanolic acid and other significant organic molecules [5,6]. A research work on the aqueous extract of *C. monogyna* harvested from the Middle Atlas region, analyzed by HPLC-UV, identified ascorbic acid, tannic acid, rosmarinic acid, gallic acid, catechin, caffeic acid, coumaric acid, and rutin [7]. Another study on the methanolic extract of *C. monogyna* collected in Spain identified isovanillic acid, kaempferol, quercitrin, ursolic acid, and apigenin. Flavonoids such as kaempferol, quercitrin, rutin, hesperetin, and arbutin were also found [8]. These differences can be attributed to the nature of the analyzed extract and the chemical analysis techniques used.

The development of new products derived from medicinal plants, harnessing both their active ingredients and secondary metabolites, represents a burgeoning scientific field in recent years [9,10]. This surge in interest stems from the potential of these herbal-based products to address limitations associated with conventional medicine, offering new avenues for treating human diseases such as cancer [11].

Leukemia, one of the most severe cancers that can result from the abnormal proliferation of white blood cells [12], is categorized into two main types: acute and chronic. This classification is primarily based on the rate of disease progression. Acute leukemia involves rapid growth and spread of abnormal blood cells, while chronic leukemia progresses more slowly. It is important to note that myeloblastic leukemia can also be further classified as acute or chronic based on the maturity of the cancerous cells [13,14].

Conventional treatment for leukemia typically involves a two-pronged approach. First, intensive chemotherapy using cytotoxic drugs aims to destroy cancerous blood cells and induce remission. This initial phase is often followed by hematopoietic stem cell transplantation (HSCT) for eligible patients. HSCT replaces the patient’s diseased bone marrow with healthy stem cells [15,16]. However, chemotherapy treatment induces severe complications for patients, including cardiac and neurological disorders [17,18]. The extensive use of these drugs can lead to chemotherapeutic drug resistance as well as gastrointestinal and liver toxicities [19].

Cancer is often associated with numerous health complications, including the activation of inflammatory and hepatotoxic pathways. This activation can result in the excessive release of inflammatory biomarkers, particularly tumor necrosis factor-alpha (TNF-α) and transforming growth factor-beta (TGF-β). Furthermore, chemotherapy is known to cause damage to both the kidneys and liver [20].

Acetaminophen overdose is known to cause liver damage. For many years, scientific research on acetaminophen overdose demonstrated its hepatotoxicity, and it has been investigated for its mechanisms with limited success in advancing therapeutic strategies [21].

The growing interest in herbal medicine stems from its potential to fight various diseases. This effectiveness is attributed to bioactive compounds within these plants, offering a range of biological and pharmacological benefits. Notably, these compounds emerge as potent anti-leukemic agents [7,16].

However, further research is still needed to fully elucidate the potential health benefits of natural bioactive compounds. Thus, this study aimed to investigate the chemical composition of *C. monogyna* hydroethanolic extract and assess its anti-inflammatory, hepatoprotective, and anticancer activities as well as the possible mechanisms of how the identified phytocompounds in the hydroethanolic extract exert the studied activities.

## 2. Results

### 2.1. Phytochemical Analysis of C. monogyna

The UHPLC-MS (Ultra-High-Performance Liquid Chromatography-Mass Spectrometry) analysis of the hydroethanolic extract of *C. monogyna*’s aerial parts (leaves and stems) revealed the presence of a diverse array of phenolic compounds, as illustrated in Figure 1. Through meticulous comparison with a lab-produced dataset (Appendix A), we were able to identify several key phenolic compounds within the extract. These include syringic acid, *p*-coumaric acid, quercetin, resveratrol, verbascoside, catechin, epicatechin, chlorogenic acid, ursolic acid, cyanidin, and quercetin.

The quantification of the identified phenolic compounds is detailed in Table 1. This table presents the concentrations of each compound, offering insight into the relative abundance of each phenolic compound within the extract. By providing a comprehensive quantification, we aim to highlight the significant presence and potential health benefits of these phenolic compounds in *C. monogyna*.

### 2.2. Cytotoxic Activity of C. monogyna

The hydroethanolic extract of *C. monogyna* exhibited significant anticancer activity against HL60 and K562 cell lines. This activity was first detectable at low concentrations, starting at 3 µg/mL for HL60 cells and 6 µg/mL for K562 cells (*p* < 0.01). After 72 h of treatment, a higher concentration of 50 µg/mL of the extract significantly reduced cell viability by more than 50% in both cell lines. The IC_50_ was 17.34 µg/mL for K562 cells and 24.08 µg/mL for HL60 cells following the 72 h treatment (Figure 2).

### 2.3. Anti-Inflammatory Activity of C. monogyna

The anti-inflammatory potential of the hydroethanolic extract of *C. monogyna* was examined by using indomethacin and NaCl as controls at 3, 4, 5, and 6 h after the carrageenan injection (0.5%). The hydroethanolic extract of *C. monogyna* at doses of 300 and 1000 mg/kg exerted a strong inhibitory effect, which reached the levels of 76.58 % and 84.12 %, respectively, after 6 h of the treatment (Table 2).

### 2.4. Hepatoprotective Effect of C. monogyna

#### 2.4.1. General Signs and Body Weight

In this investigation, the potential hepatoprotective effects of *C. monogyna* were examined. The plant extract was administered orally at 300 mg/kg and 1000 mg/kg. Compared to the negative control group, a consistent reduction in body weight was observed in animals treated with the hydroethanolic extract at 1000 mg/kg. For the positive control group as well as the group treated with the hydroethanolic extract at 300 mg/kg, no significant change in weight was observed (Figure 3).

#### 2.4.2. Relative Organ Weight

The relative organ weight (ROW) of livers, kidneys, and spleens in the context of hepatoprotective activity is displayed in Table 3. Compared to the negative control group, the oral administration of acetaminophen to animals of the positive control group increased the relative weight of the liver and kidneys. Concerning the treatment with *C. monogyna* extract at 300 mg/kg, the results did not reveal any significant change in the relative weights of the liver, kidneys, and spleens. However, the hydroethanolic extract at 1000 mg/kg showed a reduction in the relative weight of the liver.

#### 2.4.3. Biochemical Parameters

The treatment of mice with the hydroethanolic extract at a dose of 300 mg/kg and 1000 mg/kg and acetaminophen at a dose of 1000 mg/kg showed no significant increase in aminotransferases compared to the negative control group. However, the ALT (Alaninine aminotransferase) and AST (Aspartate aminotransferase) values of the mice group treated with acetaminophen alone were found to be significantly higher. In contrast, no significant changes were obtained for urea and creat (Creatinine) parameters for all tested groups (Figure 4).

### 2.5. In Silico Study

In the study of anti-inflammatory activity, procyanidin was found to be the most active molecule against lipoxygenase with a glide gscore, glide emodel, and glide energy of −7.27, −86.349, and −59.31 kcal/mol. Quercetin was the most active molecule in hepatoprotective activities with a glide gscore of −8.102 kcal/mol. Regarding anticancer activity, catechin was the most active anti-leukemia molecule against tyrosine kinase with a glide gscore of −9.037 kcal/mol, while procyanidin was the most active anti-hepatocellular carcinoma molecule against TRADD with a glide score of −8.026 kcal/mol (Table 4).

Figure 5 and Figure 6 show the 2D and 3D visualizations of the ligands in the active sites. Procyanidin was forming six hydrogen bonds in the active site of lipoxygenase with residues Ile 406, ASN 554, LEU 607, ALA 606, ALA 672, and VAL 671. In addition, in the active site of human cytochrome P450 2E1, quercetin established a single hydrogen bond with residue ASN 206 and two Pi–Pi stacking bonds with PHE 478 and HEM 500 residues.

Catechin was established by four hydrogen bonds with GLU 286, GLU 316, MET 318, and ASP 381 residues in the active site of tyrosine kinase. Furthermore, procyanidin was established by seven hydrogen bonds with ASP A 149, ALA B 446, ILE B 444, GLU B 441, LYS A 38, and LEU B 471 residues and and one Pi–cation bond with ARG A 66 residue, and Pi–Pi stacking bond with HIE A 65 residue in the active site of TRADD.

## 3. Discussion

*C. monogyna* is well known for its beneficial properties for human health. Various parts of the plant have been reported to exert antispasmodic, astringent, cardiotonic, hypotensive, anti-atherosclerotic diuretic, and hepatoprotective activities [22,23,24].

Chemical analysis of the hydroethanolic extract of *C. monogyna* by UHPLC revealed the presence of several phenolic compounds including syringic acid, *p*-coumaric acid, quercetin, resveratrol, verbascoside, catechin, epicatechin, chlorogenic acid, ursolic acid, cyanidin, and quercetin (Table 1). This is an expected range of identified phenolic compounds in this plant extract in line with the results of similar studies in this field. Several compounds have been identified in *C. monogyna*; these include chlorogenic acid, epicatechin, hypersaccharide, quercetin, rutin, vitexin, and procyanidins [25,26].

The hydroethanolic extract of *C. monogyna* demonstrated notable anticancer efficacy against the HL-60 and K-562 myeloleukemia cell lines (Figure 2). This anticancer effect could be linked to the different signaling pathways, including tyrosine kinases and TRADD pathways.

This study primarily focused on assessing the extract’s effect on cell viability and did not delve deeply into elucidating the underlying mechanisms of its action. Specifically, the impact of the extract on key signaling pathways such as the tyrosine kinase pathway, which plays a crucial role in cancer cell proliferation and survival, remains largely unexplored. Understanding the extract’s mode of action at the molecular level could provide valuable insights into its therapeutic potential and help guide further research efforts.

The observed anti-leukemia effect of the hydroethanolic extract in this study (Figure 2) may be attributed to the presence of its identified phytochemical compounds detected by UHPLC, which possess well-established biological properties.

Previous studies have demonstrated that receptor tyrosine kinases (RTKs) play crucial roles in regulating fundamental cellular processes, particularly in maintaining the balance between cell proliferation and apoptosis. Catechins are known to inhibit RTK expression by suppressing the activity of the extracellular signal-regulated kinase (ERK), which regulates the transcription factor Egr-1. Furthermore, catechins have the potential to complement existing cancer therapies that specifically target RTKs, offering a promising adjunctive approach in cancer treatment [27,28].

In addition, resveratrol has been reported to have an anti-leukemia effect by inducing apoptosis in T-cell leukemia Jurkat cell lines. This effect could be achieved through the recruitment of the Fas/CD95 death receptor and downstream Fas-associated death domain-containing protein (FADD), procaspase-8, and procaspase-10. Additionally, resveratrol promotes cleavage of Caspase-3, further contributing to its apoptotic effect [29].

Quercetin inhibits the propagation of human leukemia cells and triggers apoptosis via lessening the Bcl2-to-Bax ratio [30]. Furthermore, this flavonoid compound suppresses the expression of the angiogenesis-associated proteins HIF1α and VEGF. It also impacts the reduction of cyclin D1 protein expression and therefore blocks the cell cycle at the G1 phase [31]. In addition, quercetin exhibits significant cytotoxic effects against cancer cells, while causing little to no harm to normal cells [32].

The *C. monogyna* extract revealed an important anti-inflammatory potential, especially in the late stage of inflammation as compared with indomethacin (Table 2). This anti-inflammatory effect may be due to the phenolic compounds identified in the hydroethanolic extract of the plant.

Based on the results of our in silico study, the anti-inflammatory effect of our hydroethanolic extract can be attributed to the presence of procyanidin, verbascoside, and quercetin in the plant, as these compounds demonstrated the highest glide scores (Table 4).

The anti-inflammatory effects of *C. monogyna* have been evaluated in previous research work, through specific components such as procyanidins and triterpenes. Procyanidins, particularly procyanidin B2, have been shown to inhibit COX-1 and COX-2, crucial enzymes in the inflammatory process. These procyanidins from *C. monogyna* fruits have demonstrated significant in vitro effects against COX-2 protein expression [33]. In addition, a triterpene fraction isolated from the aerial parts (twigs, stems, and leaves) of *C. monogyna* showed promising results in vivo. At the dose of 40 mg/kg, there was a 52.5% inhibition of hind-paw edema in rats after 5 h [34].

In our study, we evaluated the hydroethanolic extract of *C. monogyna*’s anti-inflammatory effect for the first time. Our findings revealed that the hydroethanolic extract at 300 mg/kg and 1000 mg/kg doses showed strong anti-inflammatory effects, with inhibition rates of 57.66% and 79.39%, respectively, after 5 h of treatment (Table 2). Comparing our results with a previously published in vivo study on C. *monogyna* triterpene, our hydroethanolic extract provided higher inhibition percentages after 5 h of treatment. The obtained results can be attributed to the tested doses, or the used plant parts. In addition, the use of the total extract in our research might have allowed for a synergistic interaction between the various bioactive compounds, enhancing the overall anti-inflammatory efficacy.

According to other studies, verbascoside has been also reported to exert a strong anti-inflammatory activity. It may help modulate the inflammatory response by inhibiting the production of pro-inflammatory molecules like cytokines and prostaglandins, reducing the production of superoxide radicals and consequently the activity of iNOS [35,36,37].

The liver filters blood from the gastrointestinal tract, breaking down toxins and removing harmful substances from the body. Damage to the liver can result from impaired hepatic function due to alcohol, toxins, other harmful factors, and medications [38]. Acetaminophen is a well-tolerated drug at therapeutic doses, with fewer side effects. However, an overdose of acetaminophen can cause serious hepatotoxicity, in some cases progressing to liver failure and death [21]. The use of acetaminophen inducing hepatotoxicity as a model to determine the hepatoprotective effect of our plant at doses of 300 and 1000 mg/kg in mice can be considered more ethical than the use of other, more powerful hepatotoxins like CCL4.

The administration of the hydroethanolic extract at doses of 300 mg/kg and 1000 mg/kg, along with acetaminophen at a dose of 1000 mg/kg, demonstrated a potential hepatoprotective effect in mice. The tested extract did not elicit significant changes in ALT, AST, urea, and creatinine levels compared to the negative control group (Figure 4).

Several studies on *C. monogyna*’s hepatoprotective effect have already been carried out. A previous study evaluated the hepatoprotective effects of *C. monogyna* ethanolic extracts from bark, leaves, and berries collected in East and West Azerbaijan on hypercholesterolemic rats. The ethanolic extracts of hawthorn improved hypercholesterolemia-induced liver damage [39].

Another study conducted in 2020 investigated the hepatoprotective activity of aqueous extracts from the fruits and leaves of *C. monogyna*, collected from northern Algeria, against copper-sulphate-induced hepatotoxicity in rats. Copper sulphate administration induced hepatotoxicity, while the combination of this metal with the hawthorn aqueous extract attenuated the hepatotoxicity by decreasing MDA concentration and increasing the levels of antioxidants (GSH and GPx) [40].

This hepatoprotective effect can be attributed to the phenolic compounds present in the plant, particularly quercetin. Quercetin is a flavonoid known for its antioxidant properties and has been extensively researched for its potential hepatoprotective effects [41,42]. It can help neutralize harmful free radicals in the body, which can reduce oxidative stress in the liver, a common factor in liver damage and various liver diseases.

Quercetin has been shown to inhibit the activity of CYP2E1, an enzyme involved in the production of reactive oxygen species (ROS) and free radicals during the metabolism of certain substances. By inhibiting CYP2E1, quercetin may help reduce the generation of harmful ROS in the liver, leading to decreased oxidative stress and reduced liver damage [43].

The *C. monogyna* hydroethanolic extract demonstrated (i) a hepatoprotective activity against the acetaminophen-induced hepatotoxicity, and (ii) anti-inflammatory and antileukemia effects against K-562 and HL-60 cells.

## 4. Materials and Methods

### 4.1. Plant Material

Aerial parts (leaves and stems) of *C. monogyna* were harvested between March and April 2022 from the Ifrane-Azrou area. A botanist, Bari Amina, identified the plant material, and the following voucher number BPRN28 was given to the species. A sample was deposited in the herbarium of the laboratory LBEAS, Faculty of Sciences Dhar El Mehraz, Sidi Mohamed Ben Abdellah University (USMBA, Fez, Morocco). The aerial part of *C. monogyna* was dried in a shaded place in a well-ventilated room before being ground into powder.

### 4.2. Animal Material

Wistar rats used in this study, weighing between 170 and 246 g, and Swiss mice aged 2 months, weighing between 28 and 31 g, of both sexes were obtained from Fez University’s Faculty of sciences, Biological Department, under laboratory conditions, with a temperature of 23 ± 2 °C and a 12 h/dark cycle for the animals. Additionally, the mice had 24/7 access to food and water. This research followed guidelines for animal studies set by the National Research Council Committee in 2011 [44]. This study was approved by the institutional ethical committee of care and use of animals at the laboratory of Biotechnology, Health, Agrofood, and Environment of Sidi Mohamed Ben Abdallah Fez University, Morocco, (LBEAS 06/12 April 2023).

### 4.3. Preparation of the Extract

The hydroethanolic extract was obtained by mixing 10 g of *C. monogyna* leaf and stem powder with 70 mL of ethanol and 30 mL of distilled water. The extraction technique used was sonication for 45 min at 25 °C. The extraction was performed using 200 W ultrasound equipment (SB100DT, Ningbo, China). The obtained mixture was filtered through Wathman paper, then condensed using a rotary evaporator (BUCHI:461, Flawil, Switzerland) at 40 °C [45].

### 4.4. Chemical Composition

The hydroethanolic extract of *C. monogyna* was analyzed to identify the phenolic composition using ultra-high-performance liquid chromatography (Nexera, XR LC 40, Kyoto, Japan). This instrument is coupled with triple quadrupole mass spectrometry characterized by an MS/MS detector (LCMS 8060, Shimadzu, Milan, Italy); *m*/*z* ion detection was used to screen the phenolic composition of *C. monogyna* based on a database. The flow of nebulizer gas was set at 2.9 L/min, and the heater gas flow was adjusted to 10 L/min. Accordingly, the flow of the dryer gas was set at 10 L/min. In addition, the interface temperature was set to 300 °C and the DL temperature was maintained at 250 °C, and finally, we adjusted the heat block temperature to 400 °C. Concerning the MS/MS detection, it was performed using electrospray ionization. The mobile phase contained water: acetonitrile (95:5, *v*/*v*) þ formic acid (0.01%) in ESI+ and ESI–.

The separation was conducted with C18 column (Accucore Polar Premium) with the following characteristics: 2.6 mm; 46 × 150 mm, with a total running time of 5 min and flow rate of 0.7 mL/ min in isocratic conditions. *C. monogyna* extract was dissolved in 1 mL of saline phosphate buffer–acetonitrile 1:1 LC/MS grade solution for analysis [46].

### 4.5. Cytotoxicity Effect of C. monogyna Extract

The cytotoxic effects of the *C. monogyna* extract were evaluated using the methylthiazoletrazolium (MTT) assay, following a previously established protocol [47,48]. Human hepatocarcinoma (Huh-7), human acute promyelocytic leukemia (HL60: ATCC^®^ CCL-240TM), and human chronic myelogenous (K562; ATCC^®^ CCL-243TM) cell lines were cultured in 96-well plates (1 × 10^4^ cells/well). These cells were then exposed to different concentrations of the extract (ranging from 3 to 100 µg/mL) and subsequently incubated for 24, 48, and 72 h at 37 °C in a 5% CO_2_ atmosphere. Untreated cells and 100% DMSO served as negative and positive controls, respectively. After each incubation period, 10 μL of an MTT solution (5 mg/mL) was added to each well and incubated for 4 h under the same conditions as before. The reaction was halted by adding 100 μL of 0.1 NHCl in anhydrous isopropanol. Cell viability was determined by measuring absorbance using spectrophotometry with a 570 nm wavelength filter. The relative cell viability was calculated using the following formula: (absorbance at 570 nm of the treated sample)/(absorbance at 570 nm of the untreated sample) × 100. All experiments were repeated three times; therefore, results reflect triplicate measurements. These tests were conducted at the Virology and Immunology Laboratory of the National Institute of Amazonian Research in Manaus, Brazil.

### 4.6. Anti-Inflammatory Test of C. monogyna

The anti-inflammatory potential of *C. monogyna* was assessed following the protocol described previously [49]. An animal model, with 20 rats divided equally in 4 groups, was used. In the control group, we used only saline solution. The second group received the hydroethanolic extract of *C. monogyna* (300 and 1000 mg/kg). Indomethacin (10 mg/kg) was used as positive control. Intradermal injection was used to administrate 0.1 mL of carrageenan, 1 h after the oral administration of the extract and controls. The size of the paw was measured before the injection and after 3, 4, 5, and 6 h. Measurement of the circumference was performed using a cotton thread around the paw. The inhibition was expressed as percentage and calculated using the below formula:$$Inhibition(\%)=\frac{Control\;\left(St-S0\right)-Treated\;(St-S0)}{Control\;(St-S0)}\times 100$$

St = The size of the paw (mean) after carrageenan injection.

S0 = The size of the paw (mean) before carrageenan injection.

### 4.7. Hepatoprotective Effect of C. monogyna Extract

There are difficulties and obstacles that come up when using plants, mainly because they produce types of secondary metabolites, some of which have been found to be toxic [50]. For this research, an animal model was used to evaluate the safety of *C. monogyna* and its possible effects on induced liver damage.

#### 4.7.1. Experimental Protocol

Each tablet of acetaminophen was ground into a fine powder and suspended in saline solution and administered conforming to the mice body weight. Twenty animals were randomly distributed into four groups, with each group consisting of five mice. Extracts were given by gastric gavage for 7 days to animal groups as follows:Group 1: Negative control (NaCl (0.9%)).Group 2: Positive control, acetaminophen (1000 mg/kg).Group 3: Hydroethanolic extract (300 mg/kg) + acetaminophen (1000 mg/kg).Group 4: Hydroethanolic extract (1000 mg/kg) + acetaminophen (1000 mg/kg).

#### 4.7.2. Biochemical Evaluation

In accordance with guidelines for animal testing [51], the mice were administered sodium through an injection into their abdominal cavity (50 mg/kg) to induce anesthesia. The tested mice’s blood was collected using tubes containing heparin (10 U/mL heparin), then spun in a centrifuge at 1500 rpm for 10 min. Subsequently, the plasma was examined for parameters such as ALT, AST (serum transaminases), urea, and creatinine levels (Crea) [52].

#### 4.7.3. Relative Organ Weight

The treated animals’ organs were removed and weighed [46]. Each mouse’s ROW was determined as:$$ROW=\frac{Weight\;of\;the\;organ\;(g)\;}{Body\;weight\;of\;the\;animal\;(g)}\times 1000$$

ROW: Relative Organ Weight.

### 4.8. In Silico Molecular Prediction

In the molecular investigation, we examined different effects of compounds identified in *C. monogyna* extract, encompassing their effects on inflammation (lipoxygenase) and hepatoprotection (human cytochrome P450 2E1) as well as their anti-cancer properties (specifically, their antileukemic activity targeting tyrosine kinase and anti-hepatocellular carcinoma effects related to TRADD).

#### 4.8.1. Preparation of Ligands

Ligprep and Maestro were used in the procession of the ligands. This preparation procedure entailed the removal of salt residues and the addition of hydrogen atoms. Furthermore, Epik configuration was used to generates various ionization and tautomeric states at a pH range of 7 ± 2. The OPLS3e force fields were employed, and the most favorable state, considering penalties, was chosen for subsequent steps [53].

#### 4.8.2. Protein Preparation

The structures of lipoxygenase, human cytochrome P450 2E1, tyrosine kinase, and TRADD were received from the Protein Data Bank, corresponding to PDB IDs 3V99, 3LC4, 1IEP, and 1F3V, respectively [2,54,55,56].

These proteins were appropriately prepared using the protein preparation wizard in Maestro. The preparation steps included the addition of missing disulfide bonds, elimination of water molecules, incorporation of absent hydrogen atoms, completion of missing side chains, optimization of hydrogen bonding to prevent steric conflicts, and refinement of the structure through constrained minimization to attain an RMSD of 0.3 Å [53].

#### 4.8.3. Glide Standard Precision (SP) Ligand Docking

Receptor grid was established in Maestro with grid parameters by selecting an atom in the original ligand. The grid size was set at 20 Å. Van der Waals radius scaling was adjusted to 1 to mitigate nonpolar receptor regions, and a partial charge of 0.25 was applied. All ligands were subjected to docking with the target protein using the standard precision (SP) algorithm. Identifying the best-bound ligands at the active site was based on the most negative docking score [57].

### 4.9. Statistical Approaches

One-way analysis of variance (ANOVA) was performed to analyze the statistical significance of results (GraphPad 5 software, La Jolla, CA, USA). A non-linear regression analysis was conducted to determine the IC_50_ of the extract. The results were expressed as means ± standard error of the mean (SEM). *p* value < 0.05 was considered statistically significant.

## 5. Conclusions

The results of our study highlight the potential utility of the hydroethanolic extract of *C. monogyna* as a therapeutic agent for inflammation and hepatotoxicity induced by high doses of acetaminophen, as well as in leukemia. However, it is crucial to note that the efficacy of the extract is dose-dependent. At higher doses (of 1000 mg/kg or more), the hydroethanolic extract could be considered toxic.

Furthermore, in silico analysis suggests that verbascoside emerges as the most potent anti-inflammatory compound, exerting its effects through lipoxygenase inhibition. Meanwhile, quercetin demonstrates promising activity against acetaminophen-induced hepatotoxicity.

To enhance the understanding and application of these findings, it is recommended that future research delves deeper, to elucidate the mechanisms underlying the observed effects and refine therapeutic strategies. Additionally, other clinical tests of molecules purified from *C. monogyna* are requested to validate or disprove their potential use for future therapeutic strategies.

## Figures and Tables

**Figure 1 pharmaceuticals-17-00786-f001:**
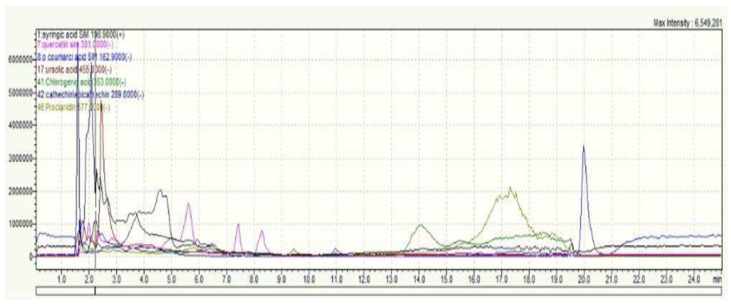
LCMS Spectrum of the identified polyphenols.

**Figure 2 pharmaceuticals-17-00786-f002:**
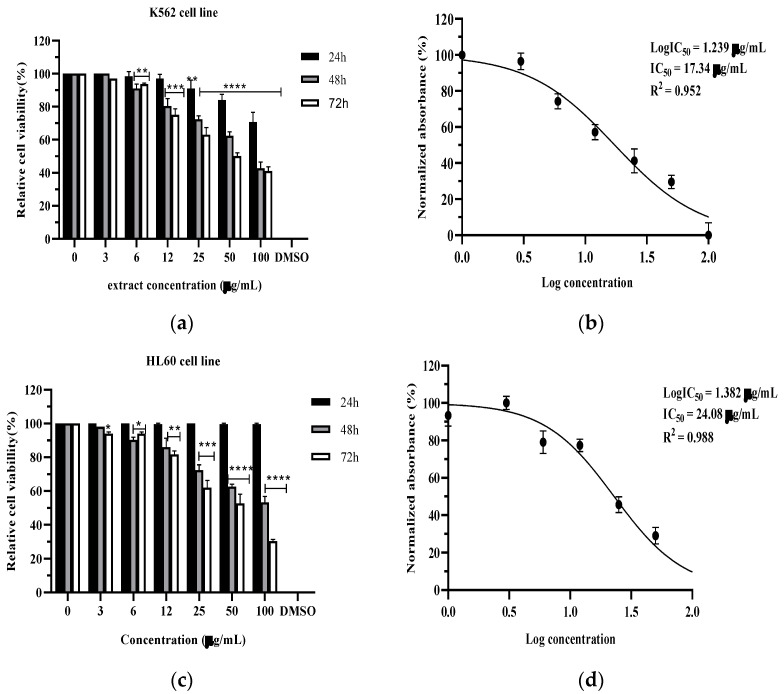
The cytotoxic effect of the *C. monogyna* hydroethanolic extract. K562 (**a**) and HL60 (**b**) cell lines were treated with different concentrations of the *C. monogyna* extract (3–100 µg/mL) for 24, 48, and 72 h time points. The IC_50_ for K562 (**c**) and HL-60 (**d**) was estimated using nonlinear regression (GraphPad Prism v. 5 software). The results are expressed as mean ± SD of three independent experiments. Statistical significance compared to the negative control (untreated cells) is denoted as follows: * *p* < 0.05, ** *p* < 0.01, *** *p* < 0.001, **** *p* < 0.0001.

**Figure 3 pharmaceuticals-17-00786-f003:**
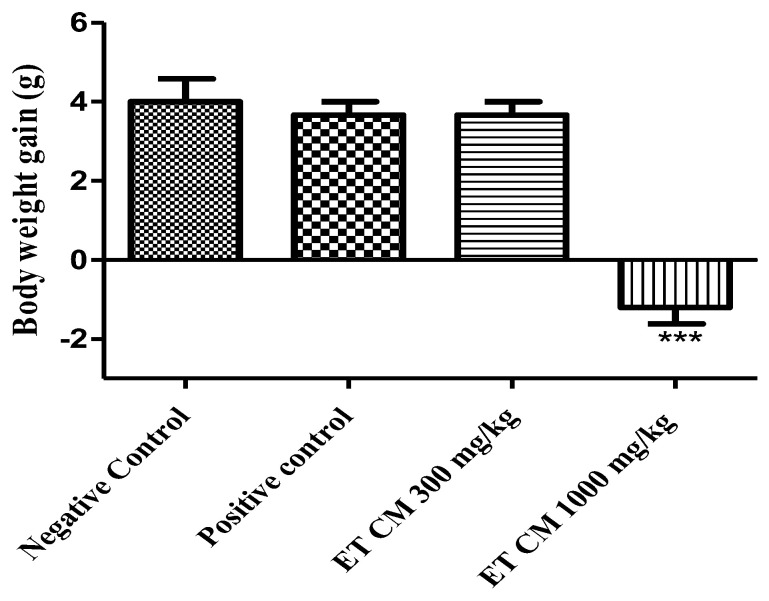
The effect of *C. monogyna* hydroethanolic extract on the body weight of mice during the hepatotoxicity test. Results are expressed as mean ± SEM (n = 5). Results are considered statistically significant compared to the negative control group (*** *p* < 0.001).

**Figure 4 pharmaceuticals-17-00786-f004:**
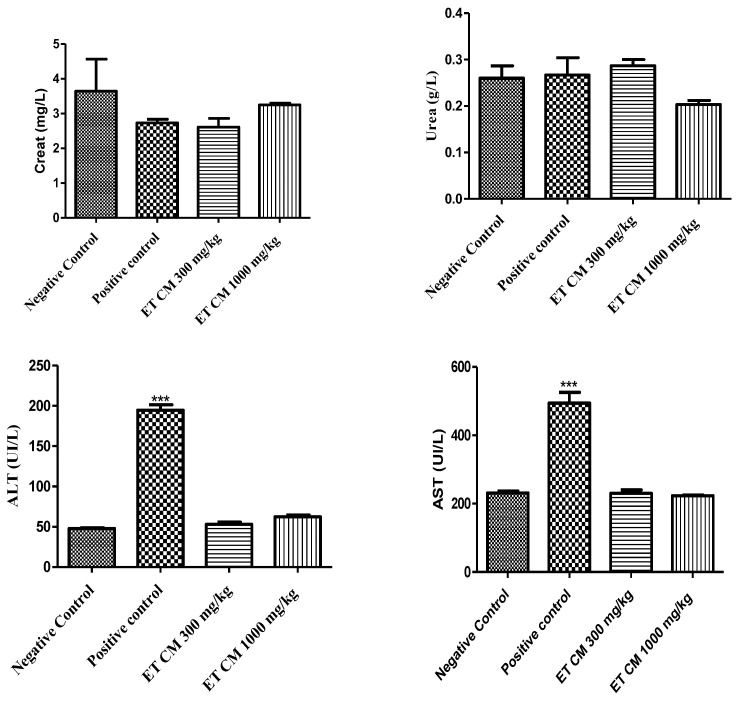
Effects of *C. monogyna* hydroethanolic extract on the liver and kidney biochemical parameters in mice. *** Indicates a significant difference at *p* < 0.001 from the control group using one-way ANOVA followed by Dunnett test. Values are expressed as mean ± SEM (n = 5).

**Figure 5 pharmaceuticals-17-00786-f005:**
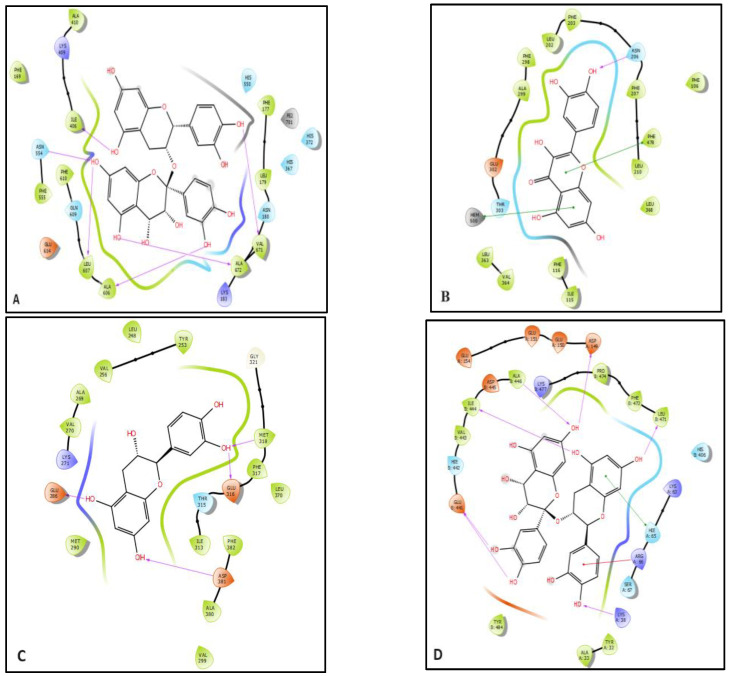
The 2D viewer of ligand interactions with the active site. (**A**,**D**): Procyanidin interactions with the active site of lipoxygenase and TRADD. (**B**): Quercetin interactions with the active site of human cytochrome P450 2E1. (**C**): Catechin interactions with the active site of tyrosine kinase.

**Figure 6 pharmaceuticals-17-00786-f006:**
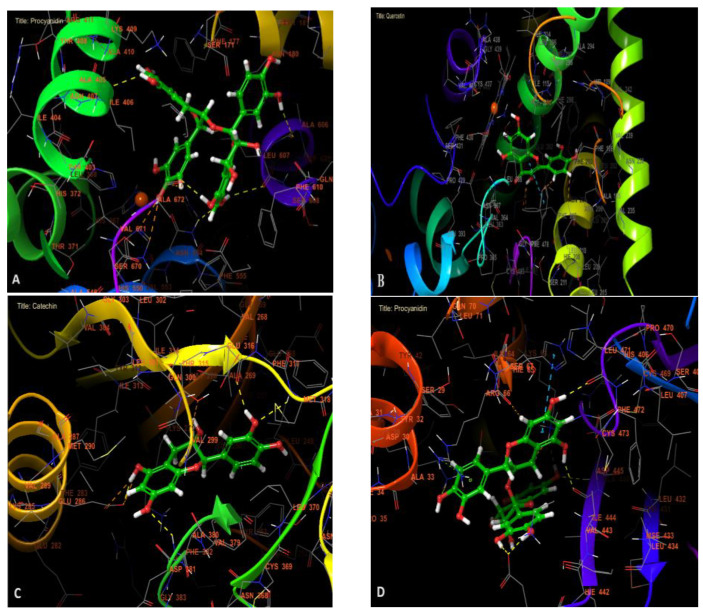
The 3D viewer of ligand interactions with the active site. (**A**,**D**): Procyanidin interactions with the active site of lipoxygenase and TRADD. (**B**): Quercetin interactions with the active site of human cytochrome P450 2E1. (**C**): Catechin interactions with the active site of tyrosine kinase.

**Table 1 pharmaceuticals-17-00786-t001:** Phenolic compounds in the extract of *C. monogyna*.

Compounds	*m*/*z*	mg/100 g
Syringic acid	197.000	123
Quercetin	302.236	171
*p*-coumaric acid	164.047	35.2
Resveratrol	228.250	353
Verbascoside	624.590	129
Catechin\Epicatehcin	289.00	458
Chlorogenic acid	353.00	415
Ursolic acid	455.00	329
Procyanidin	577.00	1356

**Table 2 pharmaceuticals-17-00786-t002:** Anti-inflammatory effects of *C. monogyna* hydroethanolic extract (ET CM) on carrageenan-induced paw edema. Values are expressed as means ± SEM. (n = 5). Results is considered statistically significant compared to the control group (* *p* < 0.05, ** *p* < 0.01, *** *p* < 0.001).

Treatment	Diameter (cm) and % of Inhibition				
	0 h	3 h	4 h	5 h	6 h
**Control NaCl (0.9%)**	2.22 ± 0.07	2.48 ± 0.09	2.51 ± 0.06	2.59 ± 0.06	2.59 ± 0.05
**Indomethacin^®^10 mg/kg**	2.25 ± 0.02	2.45 ± 0.02 * 23.08%	2.39 ± 0.02 * 50.86%	2.32 ± 0.01 *** 79.73%	2.29 ± 0.02 *** 89.19%
**ET CM 300 mg/kg**	2.30 ± 0.05	2.56 ± 0.08 * 21.79%	2.40 ± 0.07 * 42.53%	2.39 ± 0.07 ** 57.66%	2.30 ± 0.05 *** 76.58%
**ET CM 1000 mg/kg**	2.43 ± 0.07	2.60 ± 0.07 ** 32.21%	2.53 ± 0.07 ** 64.22%	2.50 ± 0.07 *** 79.39%	2.49 ± 0.07 *** 84.12%

**Table 3 pharmaceuticals-17-00786-t003:** Relative organ weights of mice change due to the administration of *C. monogyna* extract. Results are expressed as mean ± SEM (n = 5). Results are considered statistically significant compared to the negative control group. * *p* < 0.05, ** *p* < 0.01.

Organ	Negative Control	Positive Control	ET CM 300 mg/kg	ET CM 1000 mg/kg
Liver	4.46 ± 0.18	5.85 ± 0.08 **	4.64 ± 0.15	3.67 ± 0.20 *
Kidney	1.11 ± 0.01	1.53 ± 0.13 *	1.45 ± 0.11	1.46 ± 0.02
Spleen	0.44 ± 0.03	0.62 ± 0.10	0.51 ± 0.05	0.46 ± 0.01

**Table 4 pharmaceuticals-17-00786-t004:** Docking results with ligands in different receptors.

Glide (Kcal/mol)												
	Gscore	Emodel	Energy	Gscore	Emodel	Energy	Gscore	Emodel	Energy	Gscore	Emodel	Energy
	Lipoxygenase (PDB ID: 3V99)			Human Cytochrome P450 2E1 (PDB ID: 3LC4)			Tyrosine Kinase (PDB ID: 1IEP)			TRADD (PDB ID: 1F3V)		
Verbascoside	−6.84	−99.65	−66.227	-	-	-	−7.588	−96.653	−69.125	−6.57	−94.159	−68.309
*p*-coumaric acid	−6.394	−41.106	−19.167	−5.111	−30.176	−21.796	−7.154	−38.772	−25.712	−3.992	−27.567	−20.825
Syringic acid	−6.333	−43.066	−18.439	−5.335	−15.566	−14.302	−6.075	−28.892	−24.675	−4.794	−32.686	−24.841
Resveratrol	−5.471	−45.402	−33.758	−8.01	−22.454	−22.384	−8.781	−60.805	−39.999	−5.411	−42.269	−32.426
Quercetin	−6.707	−51.068	−27.781	−8.102	−55.336	−22.108	−8.687	−71.345	−50.224	−6.312	−60.618	−45.675
Catechin	−5.681	−53.357	−36.181	-	-	-	−9.037	−72.876	−48.08	−6.6	−58.723	−42.595
Chlorogenic acid	−6.132	−56.833	−32.473	−6.292	−20.511	−19.19	−5.188	−57.195	−47.258	−4.656	−54.38	−43.533
Ursolic acid	-	-	-	-	-	-	-	-	-	-	-	-
Procyanidin	−7.27	−86.349	−59.31	-	-	-	−7.459	−96.968	−66.703	−8.026	−95.471	−69.345

## Data Availability

All study data are included in the article.

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
