# Peer review of "Phenolic Composition of Crataegus monogyna Jacq. Extract and Its Anti-Inflammatory, Hepatoprotective, and Antileukemia Effects"

_pharmaceuticals, 2024, doi:10.3390/ph17060786_

Round 1

Reviewer 1 Report

Comments and Suggestions for Authors

The manuscript deals with the investigation of Crataegus monogyna Jacq. extract therapeutic properties and phenolic profile characterization. The topic is of interest and widely studied at the moment. The current study is another example focused on the plant from Morocco.

The manuscript corresponds to the Journal aims and scopes. Nevertheless, the scientific quality of the manuscript is insufficient. A range of important points needs comprehensve revision. The comments are listed below.

1. The title to be corrected, for example, as "Phenolic composition of Crataegus monogyna Jacq. extract and its anti-inflammative, hepatoprotective and  antileukemia effects".

2. Introduction is very poor. The phenolic composition of Crataegus monogyna Jacq. of various origin to be also presented and discussed. There are many articles published in this field.

3. The extract type to be detailed in the abstract and then in the Sections 2 and 4. Which parts were used for the extract preparation? There are no corresponsding descriptionin the whole manuscript. Section 4.1, "Aerial part of C. monogyna was harvested between March and April 2022, from the Ifrane-Azrou area". Section 4.3, "... by mixing 10 g of the powder with 70 mL of ethanol... ". Which parts of the plant were used? Is this leaves, flowers, berries, branches? The chemical profile of each type of plant material is very different. This point needs clarification.

4. Phenolic composition of Crataegus monogyna Jacq. does not make sense in current form of presentation.

a) First of all, the chemical names are incorrect both in the text of the section and in the table 1.

b) The corresponding chromatogram to be presented in the main text of the manuscript and mass-spectra to be shown in the supplementary materials.

c) The quantitative data on the phenolics contents to be presented. The peak area values are not representative as far as the sensitivity of the response of various compounds is not the same.

d) It seems strange, that only five phenolic compounds were identified. Many other phanolic components are identified in the Crataegus monogyna Jacq. extracts according to the literature data. This difference to be explained and discussed in details. The comparison to literature data is almost absent in the manuscrpt even in the section 3.  It would be good to add corresponding table with the comparison of the constituents reported in literature and in curent work.

5. The part of the data are shown as a single measurement results that is inappropriate. In particular, UPLC data, Fig. 1B and 1D (each point needs error bar, IC 50 to be presented as average value±SD or SEM), Table 2 (% of inhibition also needs ± SEM).

6. Fig. 1B and 1D, the X-axis title is incorrect. Concentration presented as square brackets means equilibrium concentration that is inapplicable in current case. Replace [] with c. Moreover, the X-axis title for all parts of the Fig 1 is better to show as cextract (μg/mL) for the parts A and C of the figure and lgcextract for the parts B and D.

7.  The number of significant digits in the data of Table 2 and 3 is too much. Two decimal places in both average value and SEM would be enough. Other digits are statistically insignificant.

8. Discussion to be enlarged (see also comment 4d). More ddep comparison to known data to be presented.

9. Institutional Review Board Statement section to be added at the end of the manuscript after Funding subsection.

Comments on the Quality of English Language

English is acceptable. Minor language revision would be enough.

Author Response

Please note that responses to reviewers are included in the attached file

Reviewer 2 Report

Comments and Suggestions for Authors

The paper “Revealing the phenolic composition of Crataegus monogyna Jacq. and its effect on inflammation, hepatotoxicity and cancer” investigates the phenolic composition, the anti-inflammatory, the hepatoprotective, and the anticancer activities of hydroethanolic extract.. These results underscore the unexplored potential of C. monogyna for future investigation in the treatment of leukemia, inflammation, and liver injury. Verbascoside appear to be the most active anti-inflammatory molecule via lipoxygenase inhibition. Quercetin could be the most active molecule against hepatoxicity. Finally, resveratrol is considered the strongest anti-leukemic molecule against tyrosine kinase.

The paper is interesting, the methodology is adequate and explicitly stated and the subject is very topical. The results and conclusions are remarkable and for this reason, I recommend the publication of this study after minor revision.

Therefore, the authors are invited to clarify the following aspects:

                     The manuscript should be checked for the possible writing errors (spaces between words, incorrect words), also according to the instructions for authors, the equipment must be completed with the country of acquisition

                     Page 3 line 103 please complete: syringic and p-coumaric acids

                     Page 3 figures 1 a and 1 c, the color between 48 h and 72 h is almost the same and it is difficult to see the difference. A suggestion is the use different marks (i.e. grids)

                     Page 6 table 4 I suggest to use a smaller font in table in order to have the words unsplitted because now is difficult to follow.

                     In Discussion section, I think it is good to compare the results received in this study to the results of other authors working on the same topic, if such data exist. The results obtained in this work are better than reported by other authors. Can you give some comparative values?

                     Page 10 please correct “Plant material”. Also, please describe the plant treatment (drying, particle size .. )

                     Line 330, subscript at CO2

                     I would suggest that the conclusions section be more extensive, including some final considerations on the novelties that this work provides with respect  to others already existing in the bibliography. Furthermore, it would be interesting what future work would be to delve deeper into the evidence described in this one.

Author Response

Please note that responses to Reviewers are provided in the attached doc

Round 2

Reviewer 1 Report

Comments and Suggestions for Authors

The revised manuscript is signficantly improved in scientific quality vs, initial version. All queries of the Reviewers were taken into account and corrections were inserted in the manuscript. The answers to the questions are meaningful and sufficient. The discussion of the data which was one of the important points to be corrected is emphasized in current version of the manuscript.

The only minor point that still exist but can be corrected on the editing step is symbol μ in the Figure 2.

The manuscript can be accepted to publication.

Comments on the Quality of English Language

English is ok, only minor points can be corrected.